∂ | **Open Peer Review** | Epidemiology | Research Article

# Nifuratel reduces *Salmonella* survival in macrophages by extracellular and intracellular antibacterial activity

Tian Xie,[1,2] Guifeng Liu,[1,2] Jiayi Ma,[1,2] Yaonan Wang,[1,2] Ran Gao,[1,2] Shizhong Geng,[1,2] Xinan Jiao,[1,2] Paul Barrow[3]

**ABSTRACT** *Salmonella* are intracellular bacterial pathogens for which, as with many of the other *Enterobacteriaceae*, antibiotic resistance is becoming an increasing problem. New antibiotics are being sought as recommended by the World Health Organization and other international institutions. These must be able to penetrate macrophages, and infect the major host cells and the *Salmonella*-containing vacuole. This study reports screening a small library of Food and Drug Administration (FDA)-approved drugs for their antibacterial effect in macrophages infected with a rapid-multiplying mutant of *Salmonella* Enteritidis. The most effective drug that was least toxic for macrophages was Nifuratel, a nitrofuran antibiotic already in use for parasitic infections. In mice, it provided 60% protection after oral infection with a lethal *S.* Enteritidis dose with reduced bacterial numbers in the tissues. It was effective against different serovars, including multidrug-resistant strains of *Salmonella* Typhimurium, and in macrophages from different host species and against *Listeria monocytogenes* and *Shigella flexneri*. It reduced IL-10 and STAT3 production in infected macrophages which should increase the inflammatory response against *Salmonella*.

**IMPORTANCE** *Salmonella* can keep long-term persistence in host's macrophages to evade cellular immune defense and antibiotic attack and exit in some condition and reinfect to cause salmonellosis again. In addition to multidrug resistance, this infection circle causes *Salmonella* clearance difficult in the host, and so there is a great need for new antibacterial agents that reduce intramacrophage *Salmonella* survival to block endogenous *Salmonella* reinfection.

**KEYWORDS** Nifuratel, intramacrophage *Salmonella*, antimicrobial drug

$S$*almonella enterica* spp. remain one of the important global enteric pathogens for humans and animals, threatening public health and economic development, especially in developing countries (1–3). This includes several of the typhoid-producing serovars with *Salmonella* Typhi affecting humans, and other serovars, including *Salmonella* Gallinarum and *Salmonella* Pullorum affecting poultry, *Salmonella* Choleraesuis causing severe disease in pigs, *Salmonella* Dublin affecting cattle, and *Salmonella* Abortusovis affecting goats and sheep (4, 5).

Zoonotic infections may arise from any of the more than 2,600 serovars, which generally cause diseases restricted to the alimentary tract of humans. They can also cause enteric disease in young food-producing animal species from which they arise. The major serovars are *S.* Typhimurium and *S.* Enteritidis although other serovars arise causing epidemics from time to time (6).

Over many decades, excessive use of chemotherapy to control both types of infection and for growth-promotion purposes in livestock has resulted in strains of *Salmonella* and other enteric bacteria, mainly *Escherichia coli*, resistant to many antibiotics (7, 8). In some cases, infections caused by multidrug-resistant (MDR) strains may be completely

Address correspondence to Shizhong Geng, gszzsg115@163.com.

The authors declare no conflict of interest.

See the funding table on p. 14.

untreatable with associated mortality (9). This has led the World Health Organization to restrict the use of critically important antibiotics to preserve their usefulness (10).

Withdrawal of antibiotics use is untenable as a strategy and will not result in a rapid return to antibiotic sensitivity as other environmental factors such as the presence of heavy metals and disinfectants can select for the presence of antibiotic-resistant plasmids (11–13). There is, thus, clearly a need to find new approaches to infection control including new antibiotics as advocated by international institutions (14–16).

Chemotherapy for gastroenteritis associated with *Salmonella* food poisoning is not recommended as essentially being unnecessary. One of the difficulties in treating the severe systemic diseases associated with typhoid and typhoid-like infections is the fact that *Salmonella* is essentially an intracellular pathogen, which means that antibiotic classes including aminoglycosides poorly penetrate the eukaryote cells and macrophages (17, 18), which are the main cells involved as the host for multiplication of *Salmonella* during infections (19, 20). In addition to the acute infection associated with extensive multiplication in organs rich in the macrophage-monocyte cell series, including the spleen and liver, acute infection involving the typhoid-producing serotypes may also be followed by persistent infection in a proportion of convalescents (21), with this probably depending on the genetic background of the host (22) and certainly on immune modulation by the pathogen (23, 24).

There, thus, remains a huge scope in searching for new chemotherapeutic agents and for exploring the application of existing drugs currently used for other purposes. The search for new drugs is problematic including long lead development time and associated high cost. The search for new molecules using high-throughput screening has not been as successful as was originally hoped (25). Investigation of drugs used for purposes other than bacterial infection control is also a potential route to finding new and useful antimicrobials.

The present study involved small-scale screening of an existing library of potential and existing FDA-approved drugs for their antibacterial effect initially on a *spiC* mutant of *S.* Enteritidis, which shows increased proliferation and short-term persistence in cultured macrophages (26). One active compound was retested *in vivo,* and its toxicity was assessed.

## RESULTS

### The screen model for drugs against *Salmonella* in macrophage

In a previous study, *S.* Enreritidis (SE) C50041Δ*spiC* with increased proliferation ability could maintain longer-term persistence in cultured macrophages compared to the wild-type strain (26); at 20 h post infection, C50041Δ*spiC* loads in macrophages by multiplicity of infection (MOI) = 100 were still enough, but C50041 almost could not recover, and thus this strain was selected in the study to facilitate the analysis of the drug effect at 20 h post infection.

### Primary assessment against intramacrophage *Salmonella* including cytotoxicity

After screening 66 compounds, 41 drugs with the ability to reduce *Salmonella* loads in macrophages were screened in first round and 28 compounds were focused in second round (Fig. 1A). Thirteen compounds, including E7, showed the numbers of SE C50041Δ*spiC* recovered from macrophages were less than $10 \times 10^5$ CFU, which were selected in contrast to the drug-free control, where $12.5 \times 10^6$ CFU were recovered (Fig. 2A). Considerable variation in the bacterial counts was observed with some compounds increasing the recovery of viable bacteria.

Twenty-eight compounds including the last 13 ones and 15 other control ones were assessed again for cell viability and cytotoxicity (Fig. 2B and C). Of these compounds, E7 showed no cytotoxicity at all at the concentration tested of 25 µM. This result combined with extensive bacterial reduction led to further investigation of compound E7, Nifuratel (NIF).

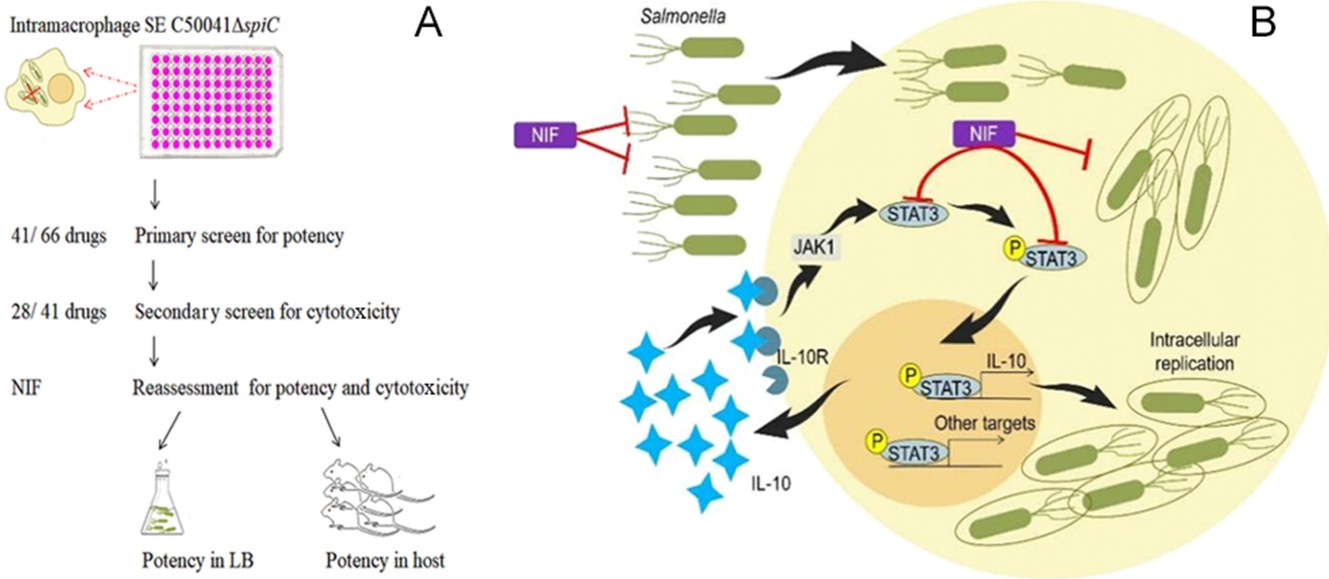

**FIG 1** NIF identification (A) and its intracellular and extracellular antibacterial effects (B).

## The effect of Nifuratel at different concentrations on bacterial numbers and cytotoxicity

NIF was compared to ciprofloxacin (CIP) for its antibacterial effect during intracellular infection (27) and also for cytotoxicity at different concentrations.

When compared to the drug-free control, Nifuratel produced a clear graded antibacterial response related to the concentration used, and at 25 µM, the antibacterial effect was higher than ciprofloxacin (Fig. 3A).

Cytotoxicity measured as a percentage (Fig. 3B) or by lactate dehydrogenase release (Fig. 3C) indicated that at the concentration that showed complete intracellular killing, 25 µM, there was very little evidence of cytotoxicity, much less than that produced by CIP. This was supported by light microscopy, which showed no obvious deterioration in the quality of the cell monolayer (Fig. 3D).

## The antibacterial effect of Nifuratel using different cell lines and bacterial strains

### Salmonella in different cell types

There was some variation in the susceptibility of SE C50041 to NIF in different cell lines. The greatest degree of susceptibility was observed in RAW264.7 cells with all bacteria killed by Nifuratel at 25 µM, whereas this occurred at 50 µM in the murine J774.1 cells, and at 100 µM in avian HD11 and human Hela cells (Fig. 4). The differences between the NIF control and at 25 µM were significant ($P < 0.01$) in all cases.

### Different Salmonella serotypes

The standard strain used here, SE C50041, was the most susceptible with all bacteria killed at a concentration of 25 µM. The *S.* Pullorum strain SPS06004 was killed completely by NIF at 100 µM, whereas complete killing was not obtained at this concentration for the *S.* Typhimurium or *S.* Dublin strains, and 25 µM had very little effect (Fig. 5).

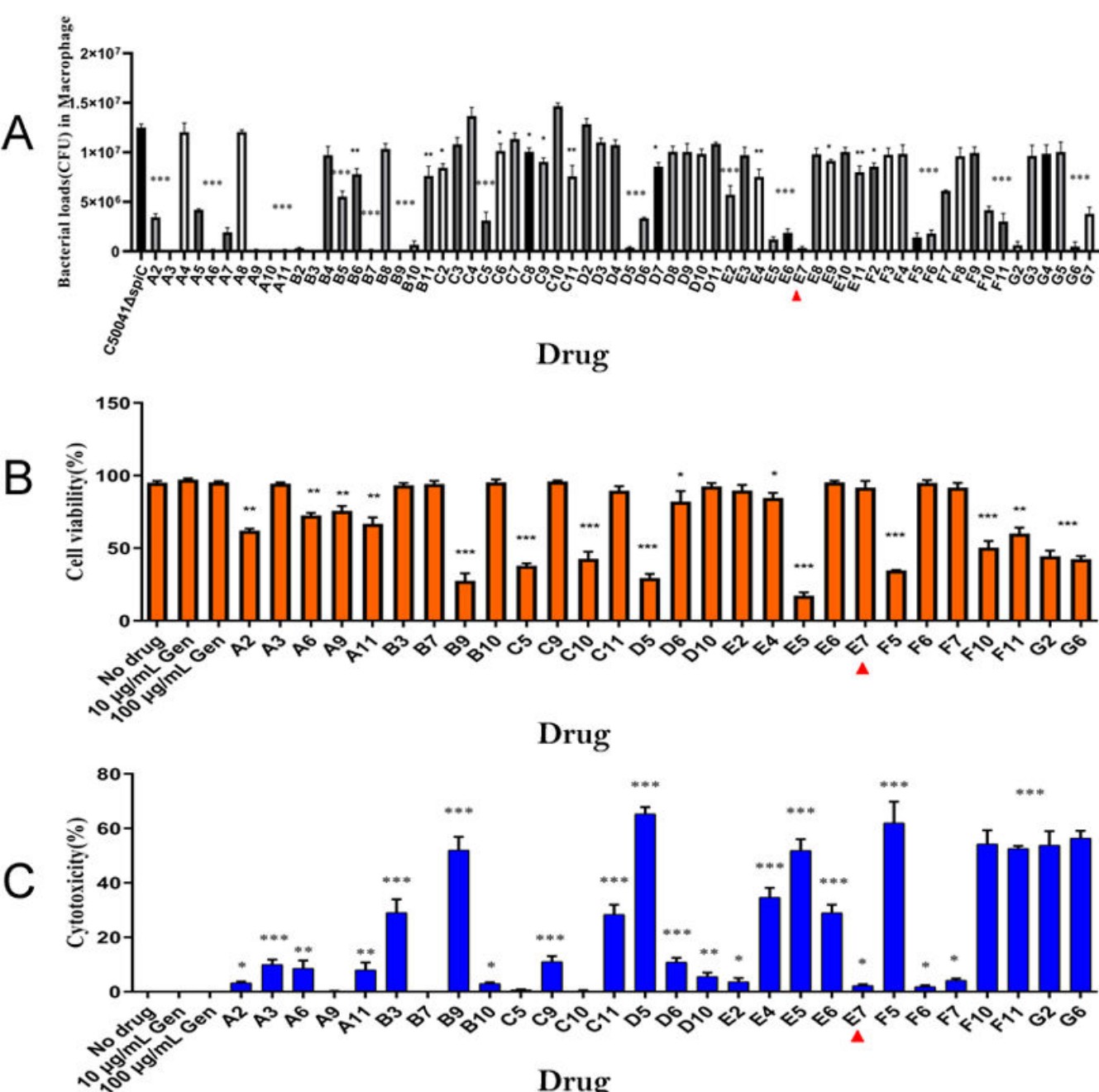

**FIG 2** Screening of drugs against intracellular *Salmonella*. Loads (CFU) of SE C50041Δ*spiC* in RAW264.7 after treatment with 66 drugs (A). Viable cell ratio (B) and cytotoxicity (C) after treatment with 28 drugs.

## Multidrug resistant Salmonella

As with the antibiotic-sensitive strain SL1344, one MDR strain of *S*. Typhimurium TX 2-7 was almost completely killed by NIF at 100 µM, whereas another MDR strain, TZF 10, was particularly resistant with no killing effect at a concentration of 25 µM, but could be effectively inhibited by Nifuratel at 100 µM (Fig. 6).

## Other intracellular pathogens

Individual strains of *Listeria monocytogenes* and *Shigella flexneri* were tested using the intracellular RAW264.7 murine cells. The *Shigella flexneri* strain L9 could also be killed

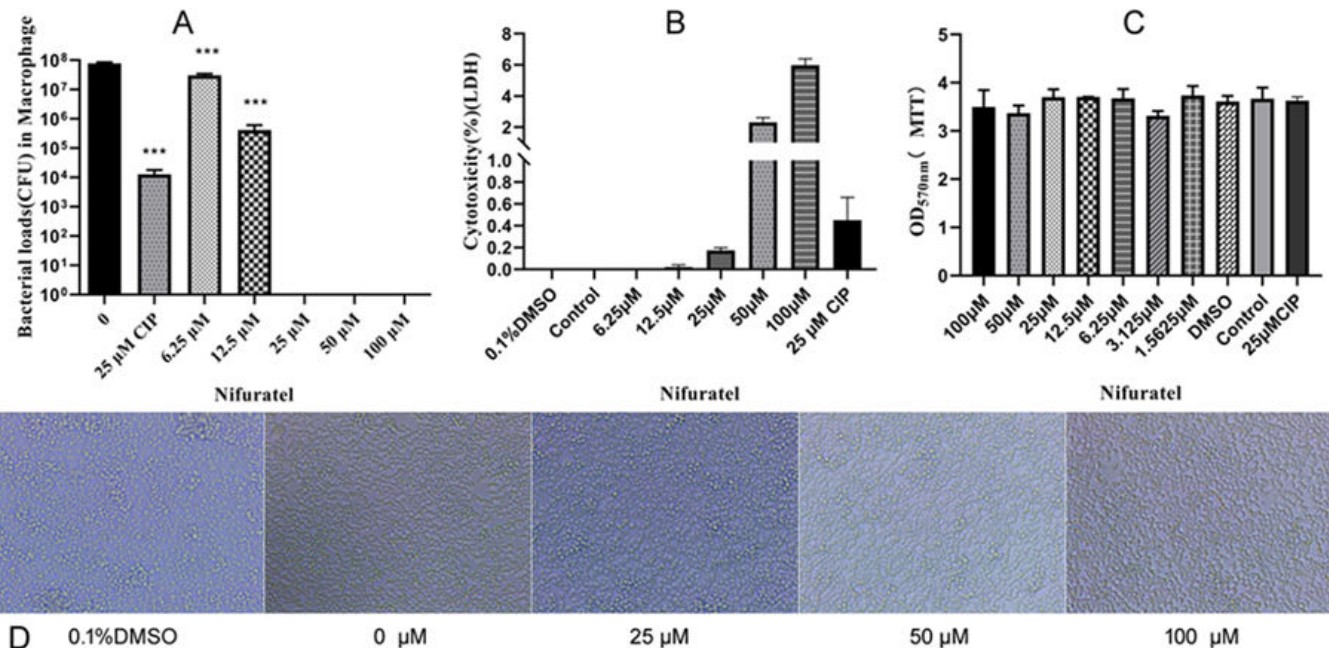

**FIG 3** The ability of NIF against intracellular *Salmonella* Enteritidis without cytotoxicity on the host cells. Loads (CFU) of SE C50041Δ*spiC* in RAW264.7 (A) LDH (B) and MTT (C) result of RAW264.7. RAW264.7 morphology after NIF treatment at different concentrations (D).

completely at 100 µM, whereas this was not the case for *Listeria monocytogenes* strain EGD-e, which showed greater resistance (Fig. 7).

## The effect of Nifuratel on the course of infection in mice infected with the parent SE C50041 strain

The viable numbers of SE C50041 in the liver and spleen of the five mice killed 48 h after oral infection with $1.0 \times 10^7$ CFU, the $LD_{100}$ of SE strain, and NIF administered (20 mg/kg of body weight) in dimethyl sulfoxide (DMSO) 12 h later are shown in Fig. 8A and B. Significant reductions were obtained in bacterial numbers for both organs ($P < 0.01$).

Mortality in mice over a 10-day period was also significantly reduced ($P < 0.01$) by Nifuratel administration (Fig. 8C). By 8-day post-infection, all non-treated mice died, whereas three mice still survived among the five treated ones.

## The effect of Nifuratel on broth cultures (extracellular) of the parent SE C50041 strain

The growth curves of SE C50041 in the presence of different concentrations of NIF are shown in Fig. 9. There was no effect on the growth rate in the logarithmic phase for concentrations up to 6.25 µM. The lag phase was increased to 12.5 and 25 µM, but the growth rate and final numbers in the stationary phase appeared similar. The lag phase

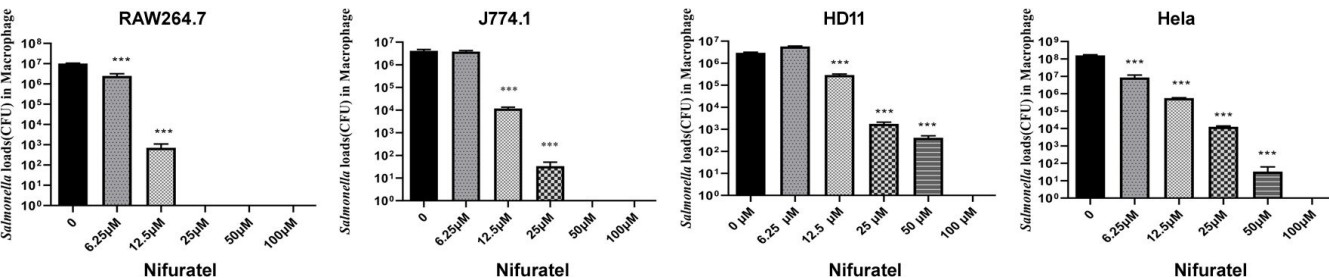

**FIG 4** The ability of NIF against intracellular *Salmonella* strain SE C50041 in different cell types (murine-derived macrophages: RAW264.7, J774A.1; avian-derived macrophages: HD-11; and human-derived epithelial cells: Hela).

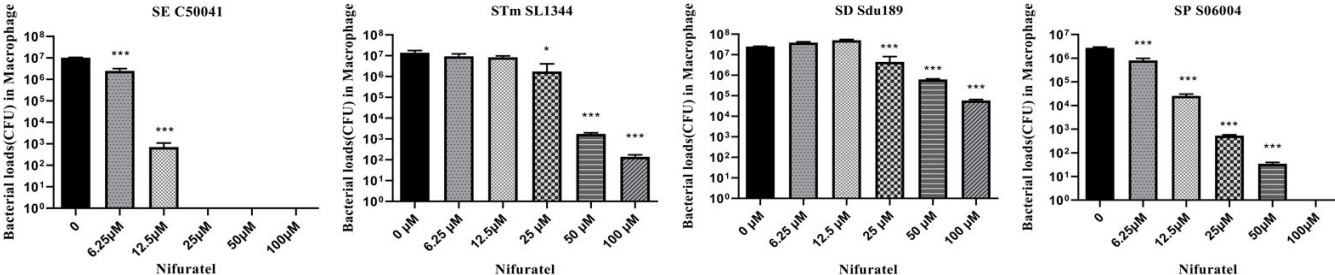

**FIG 5** The ability of NIF against different intracellular *Salmonella* serotypes.

was extended by 10 h at 50 µM, and there was no apparent growth at 100 µM. At 25 µM, there were significant differences with the untreated control at 1–10 h, $P < 0.001$; 11 h, $P < 0.01$; 12–13 h, $P < 0.05$.

## The effect of Nifuratel on IL-10 and STAT3 in macrophages infected with SE C50041Δ*spiC*

IL-10 and STAT3 production in macrophages infected with SE C50041Δ*spiC* and treated with concentrations of NIF are shown in Fig. 10. Quantitative reductions in IL-10 and STAT3 were both observed as NIF was increased in concentration (Fig. 10A and C). This was also visualized using Western blot (Fig. 10B) and when the bands were quantified (Fig. 10D), which showed that NIF could reduce IL-10 and STAT3 production in infected macrophages to increase the inflammatory response against *Salmonella* (Fig. 1B).

## DISCUSSION

The antibacterial effects of the nitrofuran molecule, NIF, were identified here by screening a small library of FDA-approved drugs for their antibacterial effects during macrophage infection by *S.* Enteritidis SE C50041Δ*spiC*.

Subsequent experiments showed that NIF was highly active against selected serovars of *Salmonella enterica*. Antibacterial activity was demonstrated during macrophage infection *in vitro*, in broth cultures, and during *in vivo* infection of mice where bacterial numbers and mortality were reduced considerably.

NIF appeared to be more antibacterial than ciprofloxacin used at the same concentration. Other drugs, not licensed for use against *Salmonella,* have also been shown to be

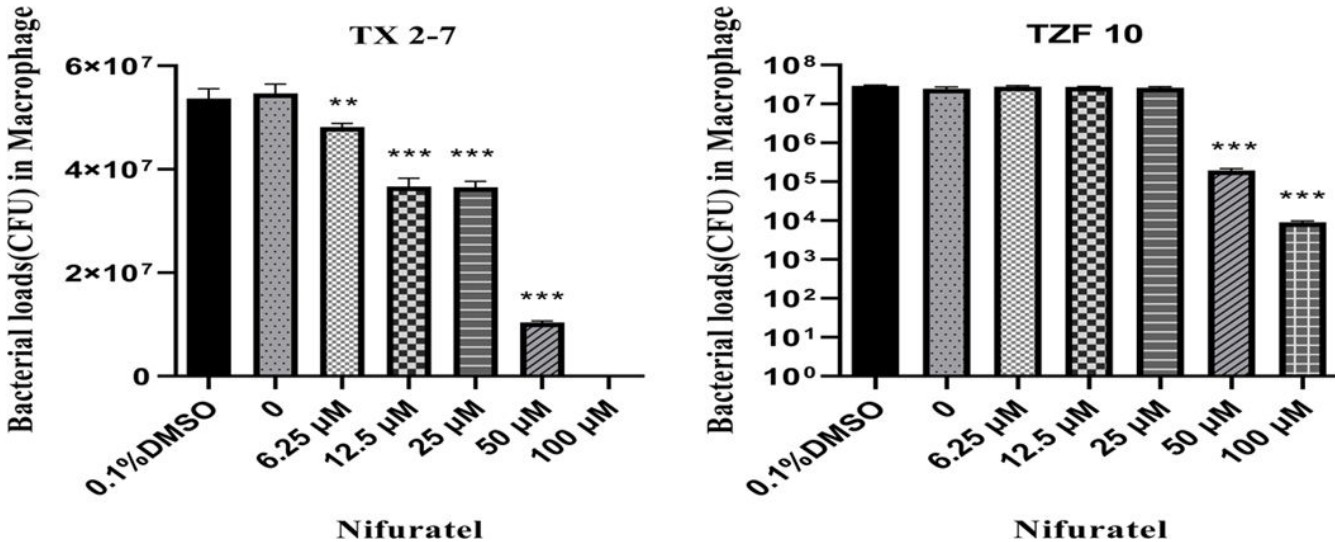

**FIG 6** The ability of NIF against intracellular multidrug-resistant *Salmonella* strains.

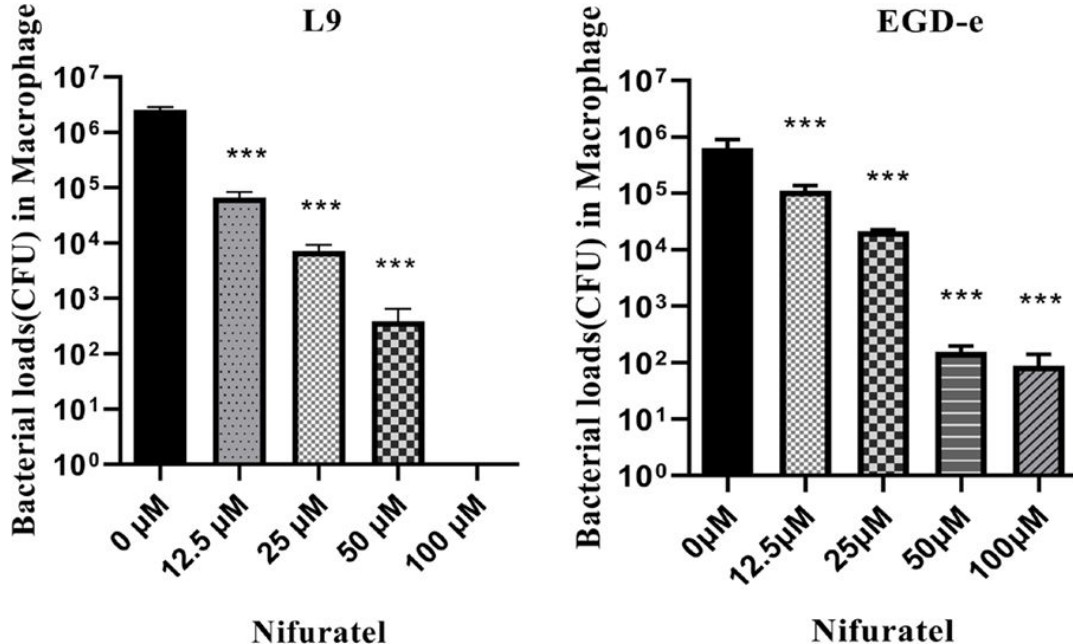

**FIG 7** The ability of NIF against intracellular Gram-negative *Shigella* L9 and Gram-positive *Listeria monocytogenes* EGD-e.

efficacious. These include Loxapine, an antipsychotic (28), Clofazimine, an anti-leprosy antimicrobial (29), Metergoline, a neuroactive drug, which is a dopamine/serotonin receptor antagonist (30). Unlike NIF, Loxapine and Metergoline are inactive against the bacteria when growing extracellularly.

At the concentration used, 25 µM, NIF showed very little cytotoxicity to macrophages as measured by the release of lactate dehydrogenase. A single dose administered orally to mice of the equivalent of 20 mg/kg of body weight was also non-toxic producing no visible adverse effects on the animals. This aligns with the current understanding of the low toxicity of NIF compared to other Nitrofuran drugs (31). NIF is rapidly absorbed after oral administration. The peak plasma concentration can be reached within 2 h, and high concentrations have been reported in saliva and vaginal secretions. Half-life is 2.75 hr. NIF is also metabolized very rapidly, and most of the oral NIF is excreted through the kidneys although it can also be excreted in urine along with two active metabolites, and unchanged drug and metabolites can also be excreted in breast milk (29, 32–38).

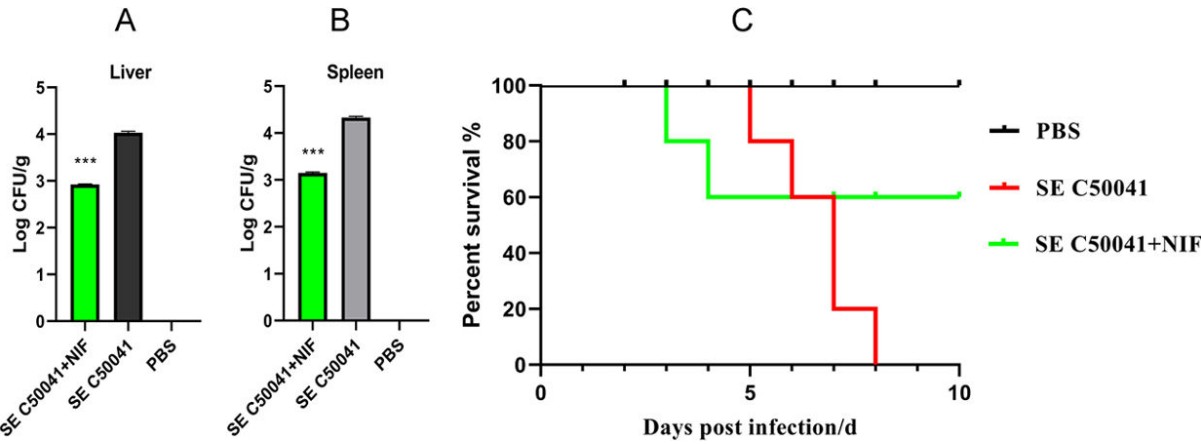

**FIG 8** NIF reduced the *Salmonella* colonization in the liver (A) and spleen (B) and mortality of mice infected with a lethal dose of *Salmonella* (C).

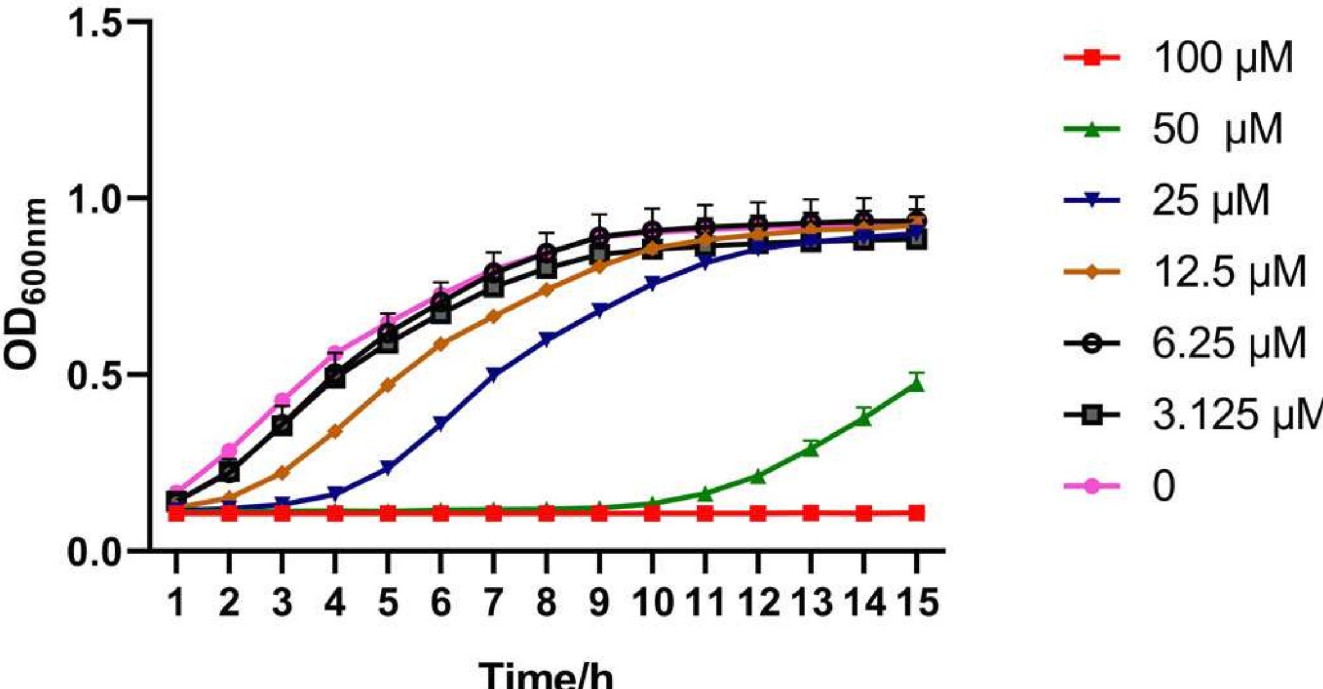

**FIG 9** Extracellular *Salmonella* strain SE C50041 survival curve after NIF treatment in different concentrations.

NIF is a nitrofuran derivative with a broad antimicrobial spectrum. It is active against the protozoan *Trichomonas vaginalis* and has an antibacterial spectrum similar to that of nitrofurantoin and some antifungal activity against *Candida albicans*. Although other drugs are preferred, NIF has been used to treat susceptible infections of the genito-urinary tract in oral doses of 200 to 400 mg three times daily. It has also been given vaginally (39).

The antibacterial effects were expressed in several macrophage cell lines from avian, murine, and human. They were also active against other *Salmonella* serovars that produce typhoid-like infections, although the greatest antibacterial effect was shown with the *S*. Enteritidis strain. It would be interesting to know how the strains used represent the serovar, given the high susceptibility of the Enteritidis strain. The two antibiotic-resistant strains of *S*. Typhimurium were also relatively resistant compared to the more sensitive Enteritidis strain. Strain TX 2.7 was comparable to the SL1344 strain, whereas the TZF 10 strain showed reductions in bacterial numbers only at 100 µM. Greater resistance was also observed in the single strains of *Listeria monocytogenes* and *Shigella Listeria monocytogenes* requiring 50 µM to have any real antibacterial effect.

By comparing the concentrations reducing *Salmonella* survival inside the cell (25 µM) and outside the cell (100 µM), NIF appears to have a greater antibacterial effect inside the cell; it was speculated that in addition to the antibacterial effect of the drug itself, it might stimulate macrophages to increased antibacterial effects against *Salmonella*.

The reduced levels of IL-10 and STAT3 produced may coincide with the reduced levels of metabolically active bacteria within the cells. However, it is also possible that NIF may reduce IL-10 level by inhibiting the activation of the host cell STAT3 pathway directly to reduce *Salmonella* intramacrophage proliferation, but whether other intracellular antibacterial mechanisms are expressed in this way requires a more in-depth study. Activation of the STAT3 pathway can induce cells to produce IL-10 beneficial to *Salmonella* survival (40). Based on the results presented here, we speculate that by inhibiting the expression of STAT3 and p-STAT3, NIF indirectly reduces the level of

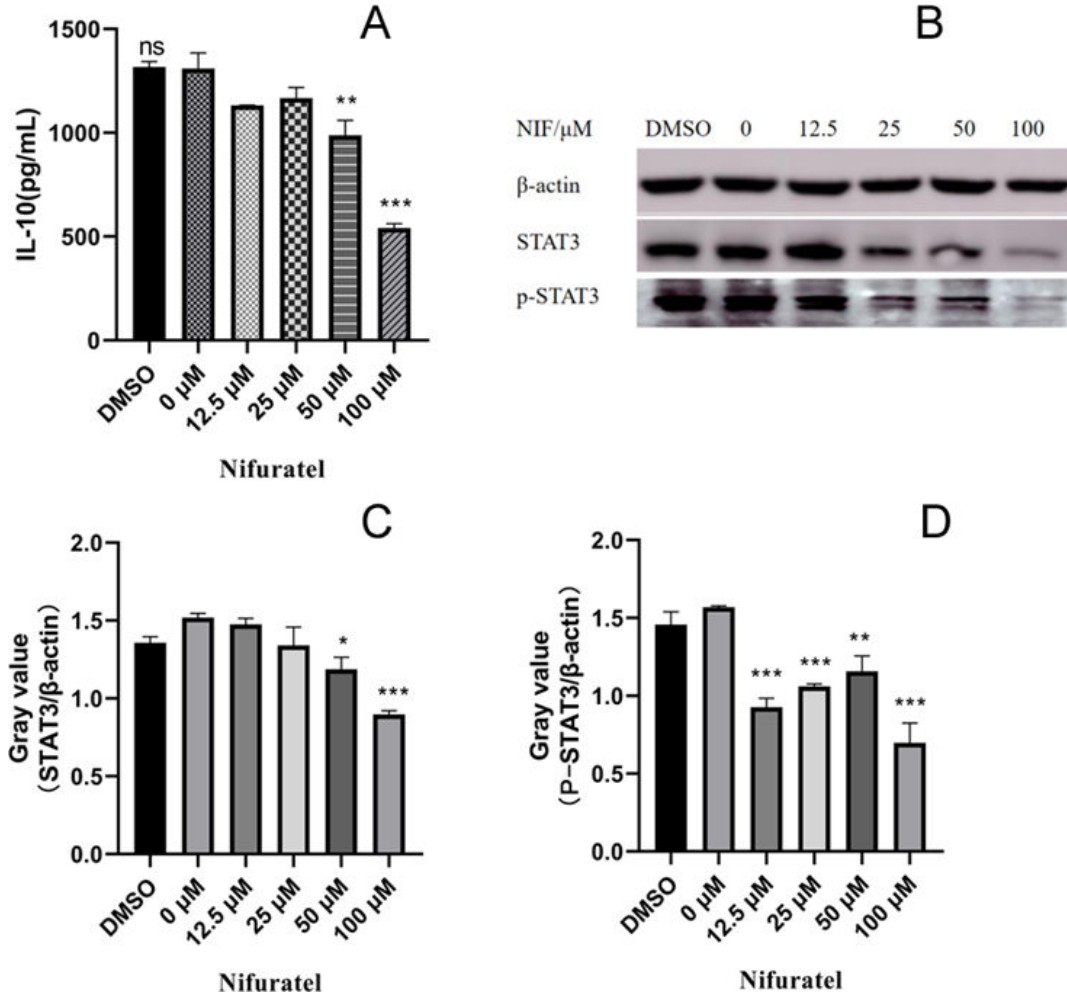

**FIG 10** NIF reduced IL-10 (A) and STAT3 level (B–D) in *Salmonella*-infected macrophages.

IL-10 and promotes the inflammatory response, thereby reducing the *Salmonella* load in macrophages (41). This requires further testing.

*In vitro*, in broth cultures, the greatest effect appeared to be in determining the length of the lag phase. What this means in terms of any differences in its mode of action compared to other nitrofurans remains to be seen since it appears to be bactericidal in its activity *in vivo* as are comparable nitrofuran drugs. Without further rigorous experiments, this would be difficult to state at this stage since, *in vivo* and in macrophages, damage of any sort to the bacteria could increase the bacterial killing effect of the cells and whole animal as indicated above.

The nitrofuran drugs used most extensively, furazolidone and nitrofurantoin, are both bactericidal, active through interference with DNA replication and protein production (42), and it is likely that NIF behaves similarly given the very similar molecular structure (43). Furazolidone was used extensively against Gram-positive and Gram-negative bacterial and parasitic infections, and it is useful against increasingly antibiotic-resistant bacteria because resistance, when it develops, is the result of chromosomal mutations, which are very rare and difficult to produce under laboratory conditions. It was used extensively in poultry rearing against Gram-negative sepsis and as a coccidiostat. It is toxic at normal clinical levels of administration and has been withdrawn from use in different countries in the 1990s. The other nitrofuran used extensively, Nitrofurantoin, remains in use for urinary tract infections but also has a range of toxicities. NIF is used in human medicine primarily for infections of the female urino-genital tract and has

been shown to have a relatively wide range of antibacterial activity with activity against *Chlamydia*, *Trichomonas vaginalis,* and *Candida* strains. It has also been shown to have low toxicity (35); at the same time, it does not affect the normal flora of *Lactobacilli* (36). It is also active against other bacterial species (37, 38).

The prevalence of MDR strains of enteric bacteria, such as *Salmonella*, *E. coli*, *Proteu*s, *Klebsiella,* and other <u>E</u>nterococcus faecium, <u>S</u>taphylococcus aureus, <u>K</u>lebsiella pneumoniae, <u>A</u>cinetobacter baumannii, <u>P</u>seudomonas aeruginosa, and <u>E</u>nterobacter species (ESKAPE) bacterial pathogens, continues to increase, and the increasing use of the fluoroquinolones has not broken this trend, with increasing resistance to these drugs due to the presence of resistance genes on both chromosomes and plasmids (44, 45). The use of a relatively underused drug such as NIF, which is highly active *in vivo*, clearly enters eucaryotic cells, unlike antibiotics such as the aminoglycosides, and shows low toxicity must be considered. Further studies on the frequency of resistance development are also advocated.

## MATERIALS AND METHODS

### Reagents

Stocks (10 mM) of the compound library of FDA-approved drugs (Topscience, TargetMol No. L4500, Shanghai, China) (Table S1) used for screening were stored at −20°C in DMSO. NIF (99.83%, Topscience, T1526, Shanghai, China) were diluted into DMSO at 100 mM and stored at −70°C. CIP (>99%, Meilunbio, MB1283, Dalian, China) was diluted into 0.2% sterile acetic acid at 100 mM and stored at −20°C (46).

### Bacterial strains and cell lines

The bacterial strains and cell lines used in this study are listed in Table 1.

### Primary assessment against intramacrophage *Salmonella*

Unless otherwise stated, prior to experiments, bacteria were grown overnight in Luria-Bertani (LB) broth overnight at 37°C with shaking at 180 rpm. RAW264.7 cells (a total of $1.0 \times 10^6$ CFU) seeded in a 24-well plate, cultured as described previously (47) for 10 h , were infected by SE C50041ΔspiC at an MOI of 100. Plates were centrifuged at 1000 ×

**TABLE 1** Bacterial strains and cells used in this study

| Strains or cells | Source | Characteristics |
|---|---|---|
| Strains | | |
| STm SL1344 | Stored in our laboratory | Wild-type *Salmonella* Typhimurium |
| SE C50041 | | Wild-type *Salmonella* Enteritidis |
| SD Sdu189 | | Wild-type *Salmonella* Dublin |
| SP S06004 | | Wild-type *Salmonella* Pullorum |
| STm SL1344ΔspiC | Constructed and stored in our laboratory | *spiC* mutant of *Salmonella* Typhimurium |
| SE C50041ΔspiC | | *spiC* mutant of *Salmonella* Enteritidis |
| SD Sdu189ΔspiC | | *spiC* mutant of *Salmonella* Dublin |
| SP S06004ΔspiC | | *spiC* mutant of *Salmonella* Pullorum |
| TX 2-7 | Isolated and stored in our laboratory | MDR *Salmonella* Typhimurium isolates |
| TZF 10 | | MDR *Salmonella* Typhimurium isolates |
| L9 | | *Shigella* |
| EGD-e | Stored in our laboratory | *Listeria monocytogenes* |
| Cells | | |
| RAW264.7 | Stored in our laboratory | Murine macrophage |
| J774A.1 | | Murine macrophage |
| HD-11 | | Avian macrophage |
| Hela | | Human epithelial cell |

g for 10 min to synchronize infection. After 30 min, the cells were washed, exposed to 100 µg/mL gentamicin for 1 h to kill non-invaded bacteria, and this was then replaced by 10 µg/mL gentamicin alone or combined with the compound to be screened from the compound library at a concentration of 50 µM; at the same time, no drug treatment was used as blank control. This time was noted as $T_0$. After 20 h ($T_{20}$), the infected cells were washed, lysed with 0.2% vol/vol Triton X-100 for 10 min at 37°C, and then serially diluted in PBS. The dilutions were spread on LB agar plates and incubated at 37°C for 18 h. The bacterial colonies grown on plates were counted and expressed as CFU.

## The effect of selected compounds on cell cytotoxicity and viability

Compounds that showed evidence of reducing intramacrophage SE C50041Δ*spiC* numbers were tested for cytotoxicity for the RAW264.7 cells. RAW264.7 macrophages ($2.0 \times 10^5$ CFU in 500 µL) were seeded into 48-well plates in Dulbecco's modified eagle medium (DMEM) with 10% fetal bovine serum (FBS) and incubated at 37°C with 5% $CO_2$ for 16 h. Compounds to be tested were premixed into Opti-MEM at a final concentration of 50 µM, and then added to wells. After 20 h of exposure, the cell culture supernatant was collected to analyze the release of lactate dehydrogenase. Absorbance was measured at 490 nm after mixing 120 µL cell culture supernatant with 60 µL lactate dehydrogenase (LDH) assay working solution for 30 min at room temperature and protected from light (48). Positive control wells were treated with 10× lysis buffer for 1 h. Percent cytotoxicity was calculated with the following formula:

$$\text{Cytotoxicity} = \frac{\text{OD}_{490\,\text{Drug treated}} - \text{OD}_{490\,\text{Spontaneous}}}{\text{OD}_{490\,\text{Maximum}} - \text{OD}_{490\,\text{Spontaneous}}} \times 100\%$$

where $\text{OD}_{490\,\text{Spontaneous}}$ is the amount of absorbance at 490 nm in the supernatant of negative control wells, $\text{OD490}_{\text{Drug treated}}$ is the amount of absorbance at 490 nm in the supernatant of sample wells, and $\text{OD}_{490\,\text{Maximum}}$ is the amount of absorbance at 490 nm in the supernatant of positive control wells. Meanwhile, the drug-treated cells were digested with 200 µL trypsin for 1 min, resuspended in DMEM medium containing 10% FBS, and subsequently the cells in the wells were counted using a cell counter and the percentage of viable cells was calculated. We retested compounds of interest at different concentrations analyzing their ability to reduce intracellular *Salmonella* numbers.

Meanwhile, a cell viability assay was performed. RAW264.7 macrophages (about $1.0 \times 10^5$ CFU in 200 µL) were seeded in 96-well plates and incubated for 16 h followed by treatment with NIF for 3 h, comparing the effect to that of CIP. Then, the cells were treated with 5 mg/mL MTT [3-(4,5-dimethylthiazol-2-yl)−2,5-diphenyltetrazolium bromide] (49) and incubated at 37°C with 5% CO2 for 1 h. The culture media were removed, and the reduced MTT products were dissolved in Formazan as a solvent. Absorbance at 570 nm was measured using a microplate reader. RAW264.7 macrophages (a total of $5.0 \times 10^5$ CFU) were seeded in 48- well plates and treated with NIF at 37°C with 5% $CO_2$ for 20 h, and the cytotoxicity was measured by using the LDH Cytotoxicity Assay Kit (Beyotime, Nantong, China). At the same time, in another assay, the morphology of Nifuratel-treated RAW264.7 cells was also observed by using electron microscopy.

## Measurement of antibacterial effect of Nifuratel against *Salmonella* strains during intracellular infection

### *Salmonella in different cell types*

In order to determine whether NIF was able to reduce intracellular *Salmonella* numbers in different cell types, four different kinds of cells were selected for the experiments using SE C50041. Infected murine-derived macrophages RAW264.7 and J774A.1, the avian-derived macrophage line HD11, and the human epithelial cell line, Hela, were infected at an MOI of 100, then treated with different concentrations (100 µM, 50 µM,

**TABLE 2** MIC of NIF to different serotypes of *Salmonella* strains

| Strains | MIC (µM) |
|---|---|
| S06004 | 50 |
| C50041 | 100 |
| SL1344 | 200 |
| Sdu189 | 200 |
| TX 2-7 | 200 |
| TZF 10 | 400 |

25 µM, 12.5 µM, 6.25 µM, and 0 µM) of NIF. After 20 h, the intracellular *Salmonella* were counted.

### Different Salmonella serotypes

RAW264.7 cells were infected by SE C50041, *S.* Typhimurium SL1344 (STm), *S.* Dublin 189 (SD), and *S.* Pullorum S06004 (SP) at an MOI of 100 and treated with different concentrations of NIF (100 µM, 50 µM, 25 µM, 12.5 µM, 6.25 µM, and 0 µM) and intracellular *Salmonella* were counted after 20 h.

### Multidrug-resistant Salmonella

The minimal inhibitory concentration (MIC) of NIF for *Salmonella* strains was determined by the broth microdilution method. Briefly, overnight bacterial cultures of strains C50041, SL1344, S06004, Sdu189, and MDR *Salmonella* Typhimurium strains TX 2-7 and TZF10 in Mueller Hinton (MH) medium were inoculated into fresh MH medium to a final concentration of $5.0 \times 10^5$ CFU/mL. *Salmonella* organisms were then exposed to NIF at increasing concentrations, ranging from 6.25 to 800 µM, in triplicate in 96-well plates at 37°C for 18 h. The MIC of NIF was defined as the lowest concentration not showing visible bacterial growth (Table 2).

After the MIC assessment, NIF activity was assessed against intracellular MDR *Salmonella* Typhimurium strains TX 2-7 and TZF10, whose patterns of resistance are shown in Table 3.

RAW264.7 cells were infected by TX 2-7 and TZF 10 at an MOI of 100 and treated with different concentrations of NIF (100 µM, 50 µM, 25 µM, 12.5 µM, 6.25 µM, and 0 µM) with the intracellular *Salmonella* counted after 20 h incubation as above.

### Other intracellular pathogens

RAW264.7 cells were infected by *Listeria monocytogenes* EGD-e and *Shigella flexneri* L9 at an MOI of 100 as above, and treated with different concentrations of NIF (100 µM, 50 µM, 25 µM, 12.5 µM, and 0 µM) followed by counting intracellular pathogens after 20 h incubation on brain heart infusion and LB plates.

### Protection ability of Nifuratel administered to mice infected with a lethal dose of *Salmonella*

BALB/c mice (6 weeks of age, *n*/group = 5, 20 ± 2 g) were distributed into five groups namely, group 1—uninfected (DMSO alone, blank control), group 2—infected (DMSO alone), and group 3—infected (NIF in DMSO) for protective effect of NIF based on

**TABLE 3** Multidrug resistances of *Salmonella* strains TX 2-7 and TZF 10[a]

| Strains | β-lactams | | | | Aminoglycosides | | Chloromycetins | | Fluoroquinolones | | |
|---|---|---|---|---|---|---|---|---|---|---|---|
| | AMX | CA | AM | CTX | Km | Sm | FON | Cm | ENX | ENR | CIP |
| TX 2-7 | R | R | R | S | R | R | R | R | R | R | R |
| TZF 10 | R | R | R | R | R | S | R | R | R | R | R |

[a]AMX: amoxicillin; CA: clavulanic acid; AM: ampicillin; CTX: cefotaxime; Km: kanamycin; Sm: streptomycin; FON: florfenicol; Cm: chloramphenicol；ENX: enoxacin; ENR：enrofloxacin；CIP：ciprofloxacin

**TABLE 4**  Groups of experimental mice

| Group | Salmonella (C50041) | | NIF (in DMSO) | | Objective |
| | Dose | Way | Dose | Way | |
| --- | --- | --- | --- | --- | --- |
| 1 | PBS | Oral | DMSO | Oral | Number of live mice at day 10 post-SE |
| 2 | $1.0 \times 10^7$ CFU | | DMSO | | |
| 3 | $1.0 \times 10^7$ CFU | | 400 µg/48 h, four times | | |
| 4 | $1.0 \times 10^7$ CFU | Oral | 400 µg, at 12 h post-SE | Oral | *Salmonella* loads *in vivo* at 48 h post-NIF |
| 5 | $1.0 \times 10^7$ CFU | Oral | DMSO | | |

mice survival rate under *Salmonella* infection with a lethal dose; groups 4 and 5—NIF treatment in DMSO for *Salmonella* load analysis *in vivo* at 48 h post-NIF (Table 4, Fig. 11). Mice were inoculated by the oro-gastric route with $1.0 \times 10^7$ CFU, the LD$_{100}$, of SE C50041. Twelve hours later NIF (20 mg/kg of body weight) in DMSO was also administered by the same route to the mice in group 3. The treatment continued for next 10 days with one treatment every 2 days. Two untreated groups were inoculated orally with the same volume of DMSO. Then, the survival rate of each group ($n = 5$, total = 15) was calculated after 10 days. The experimental protocol is shown in Fig. 11.

At the same time, group 4 and group 5 were infected and treated. In group 4, 400 µg NIF was administrated orally at the 12 h post-infection of C50041, 5 infected mice with NIF treatment were immediately euthanized by CO2 asphyxiation, followed by cervical dislocation. Spleen and liver were collected aseptically, homogenized in 1 mL of PBS and, serially diluted for plating to enumerate CFU at 48 h post-NIF . In group 5, At 12 h post-infection of C50041 without NIF treatment, 5 infected mice were also euthanized for bacterial enumeration in spleen and liver after 48 h. The effect of NIF against in vivo *Salmonella* was analyzed.

## The antibacterial effect of Nifuratel against extracellular *Salmonella*

The effect of NIF on the growth and multiplication of *Salmonella* was analyzed. A culture of SE C50041 was diluted in LB broth medium or cell culture medium to a final concentration of OD$_{600\ nm}$ = 0.05 followed by exposure to different concentrations of NIF in a flat-bottom 96-well plate, each well with a volume of 200 µL. The plate was incubated at 37 °C with shaking at 180 rpm, and the bacterial growth was monitored by measuring the absorbance at 600 nm at designated times for a total of 15 h.

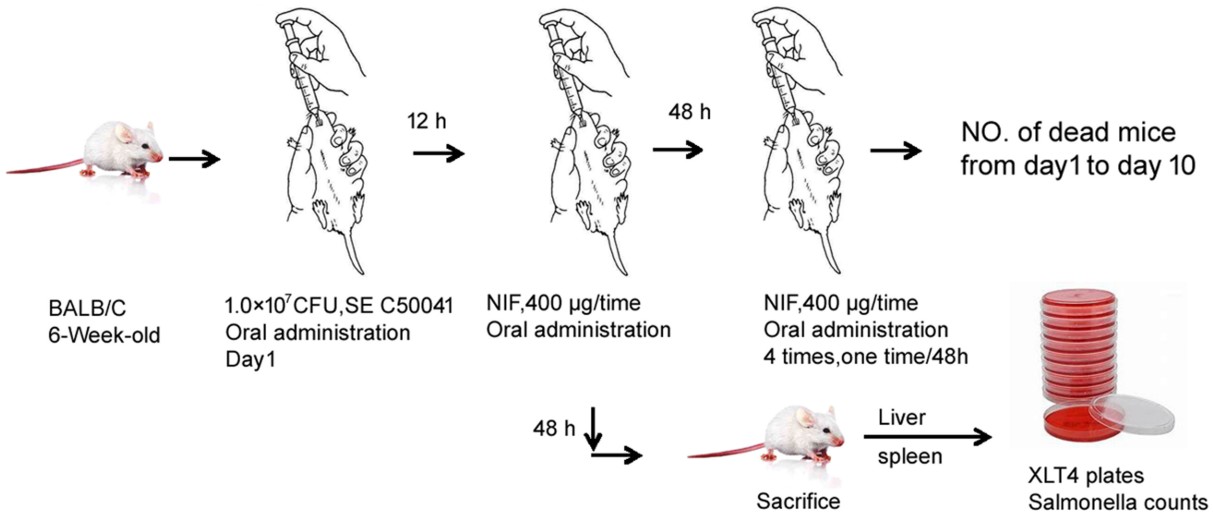

**FIG 11**  Procedures for protective efficacy of NIF in mice being infected by *Salmonella* with lethal dose.

## Analysis of IL-10 and STAT3 production in macrophages

RAW264.7 macrophages were seeded in 24-well plates and infected by SE C50041Δ*spiC* at an MOI of 100, then treated with different concentrations of NIF. Cell culture supernatants were collected 20 h later, and IL-10 levels were measured by using a Mouse IL-10 ELISA kit (BD, USA) according to the manufacturer's instructions.

Similarly, RAW264.7 macrophages were seeded in a 12-well plate and were infected by SE C50041Δ*spiC* at an MOI of 100, then treated with different concentrations of NIF. After 10 h of treatment, cells were lysed with cell lysis buffer for Western blotting (Beyotime Biotechnology, Shanghai, China). The supernatant was collected after centrifugation at $14,000 \times g$ for 5 min. The supernatant was mixed with SDS-PAGE sample loading buffer (Beyotime Biotechnology, Shanghai, China) and boiled for 10 min at 95°C. Samples were electrophoresed using SDS-PAGE and transferred onto a nitro-cellulose membrane (Pall Corporation, USA). After incubation with Phospho-STAT3 antibody (Beyotime, Nantong, China) overnight at 4°C followed by horse radish peroxidase-conjugated secondary antibody (Beyotime, China) for 1 h at room temperature, signals were measured using enhanced chemiluminescence substrate (Beyotime Biotechnology, Shanghai, China), and the image was acquired using gel documentation system. The band intensities were quantified using Image J software (ImageJ 1.50i, NIH, USA).

## Statistical analysis

The bacterial CFUs, survival, and morphometric analysis data were analyzed using GraphPad Prism 8. All experiments were repeated at least three times, and differences between group means were calculated using a two-tailed Student's *t*-test for independent samples. A confidence level of 0.05 was considered significant. In all figures, *$P < 0.05$; **$P < 0.01$; and ***$P < 0.001$.

### ACKNOWLEDGMENTS

This work was supported by the National Key Research and Development Program Special Project (2016YFD0501607), Natural Science Foundation of Jiangsu Province of China (BK20151306), Special Project on Science and Technology in North Jiangsu (SZ-SQ2021046), and Postgraduate Research & Practice Innovation Program of Jiangsu Province (SJCX22_1771).

### AUTHOR AFFILIATIONS

[1]Key Laboratory of Prevention and Control of Biological Hazard Factors (Animal Origin) for Agrifood Safety and Quality, Ministry of Agriculture of China, Yangzhou University, Yangzhou, China
[2]Key Laboratory of Zoonoses of Jiangsu Province/Co-Innovation Center for Prevention and Control of Important Animal Infectious Diseases and Zoonoses, Yangzhou University, Yangzhou, China
[3]School of Veterinary Medicine, University of Surrey, Guildford, United Kingdom

### AUTHOR ORCIDs

Tian Xie  http://orcid.org/0000-0001-7208-810X

### FUNDING

| Funder | Grant(s) | Author(s) |
| --- | --- | --- |
| Postgraduate Research & Prictice Innovation Program of Jiangsu Province | SJCX22_1771 | Tian Xie |
| MOST \| National Key Research and Development Program of China (NKPs) | 2016YFD0501607 | Shizhong Geng |

| Funder | Grant(s) | Author(s) |
|---|---|---|
| Natural Science Foundation of Jiangsu Province (Jiangsu Natural Science Foundation) | BK20151306 | Shizhong Geng |
| Special Project on Science and Technology in North Jiangsu | SZ-SQ2021046 | Shizhong Geng |

## ETHICS APPROVAL

Specified Ppathogen-Ffree (SPF) BALB/c mice (female, 6 weeks, 20 ± 2 g) were obtained from the Comparative Medical Center of Yangzhou University. All animal experiments were approved by the Animal Welfare and Ethics Committees of Yangzhou University and complied with the guidelines of the Institutional Administrative Committee and Ethics Committee of Laboratory Animals (IACUC license number: YZUDWLL-202103–083).

## ADDITIONAL FILES

The following material is available online.

### Supplemental Material

**Supplementary Table 1 (Spectrum05147-22-s0001.docx).** Information of compound library with 66 drugs.

### Open Peer Review

**PEER REVIEW HISTORY (review-history.pdf).** An accounting of the reviewer comments and feedback.

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
