## [Reviewer comments · Microbiology Spectrum]

Microbiology Spectrum

Nifuratel reduces *Salmonella* survival in macrophages by extracellular and intracellular antibacterial activity

Tian Xie, GuiFeng Liu, JiaYi Ma, YaoNan Wang, Ran Gao, Shizhong Geng, Xinan Jiao, and Paul Barrow

Corresponding Author(s): Shizhong Geng, Yangzhou University

Review Timeline:

Submission Date:	December 15, 2022
Editorial Decision:	March 26, 2023
Revision Received:	June 12, 2023
Accepted:	July 26, 2023

Editor: Cezar Khursigara

Reviewer(s): The reviewers have opted to remain anonymous.

Transaction Report:

DOI: <https://doi.org/10.1128/spectrum.05147-22>

March 26, 2023

Dr. Shizhong Geng
Yangzhou University
WenHui Street
Yangzhou
China

Re: Spectrum05147-22 (**Nifuratel reduces *Salmonella* survival in macrophages by extracellular and intracellular antibacterial activity**)

Dear Dr. Shizhong Geng:

Two experts have reviewed your manuscript and agree that it requires revisions before being considered for publication in Spectrum. Please address all of the comments when submitting a revised manuscript.

Link Not Available

Sincerely,

Cezar Khursigara

Journals Department
Reviewer comments:

Reviewer #1 (Comments for the Author):

The manuscript titled "Nifuratel reduces *Salmonella* survival in macrophages by extracellular and intracellular antibacterial activity" authored by Xie et al., has addressed pertinent problem of inhibiting *Salmonella* inhabiting within macrophages. The authors also demonstrated repurposing antiparasitic drug, Nifuratel to inhibit MDR *Salmonella*.

The manuscript warrants major revision, the English communication is extremely poor. It is very hard to understand what is written. The manuscript has been submitted with no basic formatting, no line numbers or page numbers.

Table 2: There is no possibility of determining "MIC of different *Salmonella* serotypes" . MIC Stands for Minimum inhibitory

concentration of antibiotics where no visible growth is observed. It could be MIC of compound (Nifuratel) against different strains of Salmonella. Please correct the table title.

Fig 2. Is it "Durg" or "Drug" Please clarify. Edit the typo.

The authors pretend significance level calculation by stating p value in the text. The authors may explain how many biological replicates were considered in determining significance level.

Fig. 2B does not have error bars.

The following two statements are self contradictory: "As with the antibiotic sensitive strain SL1344, the two MDR strains of S. Typhimurium TX 2-7 and TZF 10 were not killed completely by Nifuratel at 100 μ M (Fig. 6)" in the manuscript text at page # 4 and the Figure legend of the Fig. 6 " Figure 6. Nifuratel showed potent activity against intramacrophage antibiotic resistant Salmonella Typhimurium" . Please clarify.

Page #: it is not Lactose dehydrogenase, it is lactate dehydrogenase in the sentence "Cytotoxicity measured as a percentage (Fig. 3A, middle panel) or by lactose dehydrogenase release (Fig. 3A, right panel)"

Figure 3A Right panel: The OD570 seems to be identical at all the concentrations of Nifuratel as compared to control and DMSO. Are cells equally viable at all the concentration of Nifuratel? How significant your data are? Please clarify and state how Nifuratel has become less cytotoxic at 25 μ M, even better than CIP, if the cells are all equally viable in all concentrations? What was the control taken in this experiment? Ciprofloxacin (CIP) might have missed out in this experiment as it was not plotted. Please clarify the sentence on page # 5: "The antibacterial effects were expressed in several macrophage cell lines, avian, murine and human"

In Fig. 2B, F6 compound exhibited less cytotoxicity and more cell viability as compared to E7 (Nifuratel). However, the F6 has not been tested to determine intracellular bacterial load (Fig. 2A). How come authors infer that E7 could be potent candidate to serve as future therapeutic?

What are the plausible reasons of ineffectiveness of Nifuratel on antibiotic resistant Salmonella?

Can author clarify how an antiparasitic drug could be potent inhibitor of an enteric bacterial pathogen? Does the inhibitor bind and inhibit the function of paralogous enzyme(s) in both the organisms (parasite and bacteria) in the same fashion? It would be nice if authors justify this point as well in the discussion.

Reviewer #2 (Comments for the Author):

General comments:

The rationale behind these studies is strong; i.e., to identify agents with antimicrobial activity versus intra-macrophage Salmonella. Furthermore, the experiments appear to have been carefully done with appropriate methodology, and hence the data are solid. However, the clinical relevance of the study is weakened considerably by two considerations. First, as emphasized in the Introduction, the potential relevance of these studies would be to "typhoid and typhoid-like infections", but data presented is limited to non-typhoid strains, particularly Salmonella Enteritidis. Second, activity of the most promising agent, nifuratel, was disappointingly low versus all Salmonella strains except the mutant Enteritidis strain SE C50041 Δ spiC. (Relatedly, clarification is needed as to what the Δ spiC mutation is and why this strain used as model.)

With respect to manuscript preparation, the figure axis labels are inconsistent and/or incorrect, and the figure legends require more details.

A control experiment without continuous gentamicin would be useful to rule out the possibility that the compounds screened simply altered macrophage membrane permeability and hence allowed gentamicin entry.

Specific comments (no line numbering, so text to be edited in quotations):

Figure y axis labels inconsistent and incorrect; i.e., "Intracellular bacteria/CFU" in multiple figures when it should be "Intracellular bacteria (CFU)/macrophage" (or better yet, just "CFU/macrophage", with specifics including bacterial strain and macrophage cell line clearly noted in Figure legend).

Confusing, reword: "...antibiotic classes including aminoglycosides penetrate the eukaryote cells, including macrophages (17,18), poorly, the main cell involved as the host for multiplication of the Salmonella infections (19,20)."

"After the first-round screen of 66 compounds..." Insert "(Appendix)". (However, this should probably be "Supplementary Table 1" or the like rather than "Appendix")

"...10 x 10E5 CFU, and one of these, E7, was selected in contrast to the drug-free control where 12.5 x 10E6 CFU..." These numbers do not appear to match those in Fig. 2A. Specifically, the no drug control yielded ca. 125 CFU/5 x 10E5 RAW cells, so to increase this 10E5 times to 12.5 x 10E6 CFU would require 5 x 10E10 RAW, an unlikely high number.

"E7 showed no cytotoxicity at all at the concentration tested of [? insert number] μM ."

"S. Pullorum strain SPS06004 was killed completely by Nifuratel at 100 μM whereas complete killing was not obtained at this concentration for the S. Typhimurium or S. Dublin strains". Dissapointedly low activity then with all but SE C50041 ΔspiC .

"Figure 5. Nifuratel showed potent activity against intramacrophage SE C50041, STm SL1344, SD Sdu189, SP S06004." No, since as noted in text, "potent" activity only obtained versus SE C50041 mutant.

"...mice killed 48 h after oral infection with the SE strain and administered Nifuratel..." Indicate dose.

"The P values for each Nifuratel concentration and strain are shown." Not needed here; delete.

Fig 2A y axis label: Add "RAW264.7" after "5 x 10E5" (to reduce length, "Intracellular bacterial load" can be deleted).

Fig. 2B y axis label: Change to "Cell viability and cytotoxicity (%)"

Fig. 2B x axis label: Correct spelling of "Drug"

Fig. 3A y axis label for graph on left: Why is this axis (10E0 to 10E8) so different than in Fig. 2A (0 to 200)? Use consistent and correct label for this axis.

Fig. 3A legend: Descriptions needed for graphs in center and on right.

Fig. 4, Fig.5 y axis: Again, why this axis and Fig. 2A axis so different? Also again, correct y axis label and make it consistent with Fig. 2A and 3A.

Fig. 5 legend: Specify which macrophage used. Insert "Salmonella strains" before SE C50041

"Subsequent experiments showed that Nifuratel was highly active against selected serovars of Salmonella enterica." As noted above, "highly active" was only observed for SE C50041 ΔspiC

"Antibacterial activity, highly likely to be bactericidal..." Speculative since not tested - delete.

"It should not be surprising that some new antibiotics and other less conventional chemotherapies are found to be more effective than older, standard antibiotics (27). showed that highly lytic bacteriophages could be more effective than single doses of streptomycin in E. coli-infected mice and equally effective to eight doses." Vague and overly general - delete.

"Nifuratel is a nitrofurantoin derivative...It has also been given vaginally." References needed for this entire paragraph.

"Nifuratel is a good potential drug for the first-line treatment of bacterial vaginosis (35,36). It is active in vitro against the pool of bacteria recognized to cause bacterial vaginosis and, at the same time, it does not affect the normal the flora of lactobacilli (35). It is also active against other bacterial species (37,38)." Redundant with text above - delete or reduce considerably.

"Our previous studies have shown that S. Enteritidis (SE) C50041 ΔspiC had an increased ability to proliferate and persist in cultured macrophages in comparison with the wild-type strain (26). On this basis, we selected this strain for screening in RAW264.7" Move from Materials and Methods to beginning of Results.

"...replaced by 10 $\mu\text{g}/\text{mL}$ gentamicin alone or combined with the compound to be screened..." A control experiment without this continuous gentamicin would be useful to rule out the possibility that the compounds screened simply altered macrophage membrane permeability and hence allowed gentamicin entry.

"...analysis for release of lactate dehydrogenase. Absorbance was measured at 490 nm after mixing 120 μL cell culture supernatant with 60 μL LDH assay working solution..." Reference, or reagent source, or more details needed.

"Twelve hours later Nifuratel (20 mg/kg) in DMSO was also administered by the same route to the mice in group 3" According to Results, this was 24 h later ("administered Nifuratel 24h later")

Staff Comments:

Preparing Revision Guidelines

Please return the manuscript within 60 days; if you cannot complete the modification within this time period, please contact me. If you do not wish to modify the manuscript and prefer to submit it to another journal, please notify me of your decision immediately so that the manuscript may be formally withdrawn from consideration by Microbiology Spectrum.

Responses to Reviewers

Dear Editor and Reviewers,

We would like to thank the editor for hard work and the reviewers for suggestions and comments to improve our submitted manuscript (Spectrum05147-22 "**Nifuratel reduces Salmonella survival in macrophages by extracellular and intracellular antibacterial activity**"). According to these suggestions and comments, we have revised the manuscript and highlighted all the revisions in red colour, which were listed one by one as following:

Responses to Reviewer #1:

Comment 1: "The manuscript has been submitted with no basic formatting, no line numbers or page numbers."

Response: The **Line** and **page** numbers in the manuscript have been **added** according to the suggestion;

Comment 2: "Table 2: There is no possibility of determining "**MIC of different Salmonella serotypes**". MIC Stands for Minimum inhibitory concentration of antibiotics where no visible growth is observed. It could be MIC of compound (Nifuratel) against different strains of Salmonella. Please correct the table title."

Response: The title of Table 2 has been **changed** to "**MIC of NIF to different serotypes of Salmonella**";

Comment 3: "Fig 2. Is it "Durg" or "Drug" ."

Response: We are sorry for bothering you with this kind of mistakes that we should have avoided. In Fig 2, "Durg" has been **changed** to "**Drugs**";

Comment 4: "The authors pretend significance level calculation by stating p value in the text. The authors may explain how many biological replicates were considered in determining significance level."

Response: This have been done and discussed in **line 328-332**;

Comment 5: "Fig. 2B does not have error bars."

Response: Fig. 2B has been **redrawn**, and was **divided into two figures**;

Comment 6: The following two statements are self contradictory: "As with the antibiotic sensitive strain SL1344, the two MDR strains of S. Typhimurium TX 2-7 and TZF 10 were not killed completely by Nifuratel at 100 μ M (Fig. 6)" in the manuscript text at page # 4 and the Figure legend of the Fig. 6 " Figure 6. Nifuratel showed potent activity against intramacrophage antibiotic resistant Salmonella

Typhimurium" . Please clarify."

Response: The sentence of "As with the antibiotic sensitive strain SL1344, the two MDR strains of *S. Typhimurium* TX 2-7 and TZF 10 were not killed completely by Nifuratel at 100 μ M (Fig. 6)" at page # 4 has been changed to "As with the antibiotic sensitive strain SL1344, one MDR strain of *S. Typhimurium* TX 2-7 was almost completely killed by Nifuratel at 100 μ M, another MDR strain TZF 10 was particularly resistant with no killing effect at all at a concentration of 25 μ M, but could be effectively inhibited by Nifurate at 100 μ M.";

The legend of Fig. 6 "Nifuratel showed activity against intramacrophage antibiotic resistant *Salmonella* Typhimurium." has been changed to "The ability of NIF against intracellular multidrug-resistant *Salmonella*.";

Comment 7: It is not Lactose dehydrogenase, it is lactate dehydrogenase in the sentence "Cytotoxicity measured as a percentage (Fig. 3A, middle panel) or by lactose dehydrogenase release (Fig. 3A, right panel)";

Response: The "lactose" in the sentence has been changed to "lactate";

Comment 8: Figure 3A Right panel: The OD570 seems to be identical at all the concentrations of Nifuratel as compared to control and DMSO. Are cells equally viable at all the concentration of Nifuratel? How significant your data are? Please clarify and state how Nifuratel has become less cytotoxic at 25 μ M, even better than CIP, if the cells are all equally viable in all concentrations? What was the control taken in this experiment? Ciprofloxacin (CIP) might have missed out in this experiment as it was not plotted.

Response: These are very good questions, at first, it turns out that cells indeed have equally been viable at all the concentration of Nifuratel, similar studies can be found in Reference (33) . Second, CIP is a clinical antibacterial agent for intracellular *Samlmonella*, and the reference (Askoura, et al, 2020) has been added. The control in this experiment was "the blank control group" which is no drug in this experiment, at the same time, DMSO as drug solvent was also used as a control. The result of Ciprofloxacin (CIP) has been added in Figure 3A(right);

Reference: 27. Askoura M, Hegazy WAH. Ciprofloxacin interferes with *Salmonella* Typhimurium intracellular survival and host virulence through repression of *Salmonella* pathogenicity island-2 (SPI-2) genes expression. *Pathog Dis.* 2020;78(1):ftaa011

Comment 9: Please clarify the sentence on page # 5: "The antibacterial effects were expressed in several macrophage cell lines, avian, murine and human"

Response: The sentence has been **changed** to "The antibacterial effects were expressed in several macrophage cell lines **came from** avian, murine and human";

Comment 10: In Fig 2B, F6 compound exhibited less cytotoxicity and more cell viability as compared to E7 (Nifuratel). However, the F6 has not been tested to determine intracellular bacterial load (Fig. 2A). How come authors infer that E7 could be potent candidate to serve as future therapeutic?

Response: That's a good question, I choose **E7** as one **targeted drug** based on the drug properties with low cytotoxicity, high cell viability and high antibacterial effect, and this drug was rarely studied against intracellular *Salmonella*; but F6(Pefloxacin mesylate dihydrate, which is same with CIP as Quinolone antibiotic) has been studied.

Comment 11: What are the plausible reasons of **ineffectiveness** of Nifuratel on antibiotic resistant *Salmonella*?

Can author clarify how an antiparasitic drug could be potent inhibitor of an enteric bacterial pathogen? Does the inhibitor bind and inhibit the function of paralogous enzyme(s) in both the organisms (parasite and bacteria) in the same fashion? It would be nice if authors justify this point as well in the discussion.

Response: **I am sorry**, Nifuratel **is effective to** antibiotic-resistant *Salmonella*, such as TX 2-7, TZF 10 need higher concentration. Due to the detailed mechanism, but it is a good question, we mainly focus on this phenotype, we also worked on this mechanism by inhibiting STAT3 phosphorylation to promote inflammatory response to kill intramacrophage *Salmonella* in the paper, we will study further in-depth.

Responses to Reviewer #2

General comments:

Comment 1: First, as emphasized in the Introduction, the potential relevance of these studies would be to "typhoid and typhoid-like infections", but data presented is limited to non-typhoid strains, particularly *Salmonella* Enteritidis.

Response: **Many** *Salmonella* serotypes have been done, among them, *S. Typhimurium* SL1344, (STm) is **typhoid** strains, TX 2-7 and TZF 10 are *S. Typhimurium*. *S. Pullorum* is **typhoid strain to chicken**. Other strains such as SE C50041 and *S. Dublin* 189 (SD) are **non-typhoid** strains. We will test more strains in the future;

Comment 2: Second, activity of the most promising agent, nifuratel, was disappointingly low versus all *Salmonella* strains except the mutant Enteritidis strain SE C50041 Δ *spiC*. (Relatedly, clarification is needed as to what the Δ *spiC* mutation is and why this strain used as model.)

Response: At first, "SE C50041ΔspiC"(line 95,100) in the manuscript have been modified to "SE C50041". Second, Nifuratel is effective to tested *Salmonella* strains, but the degree is different. Our previous studies(Wang, et al, 2021) have shown that C50041Δ*spiC* keep persistence with increased ability to proliferate in cultured macrophages in comparison with the wild-type strain, and so this strain was selected for durg screen in RAW264.7 to facilitate the analysis of the effect at 20 h poi;

Reference: 26. Wang Y, Liu G, Zhang J, Gu D, Hu M, Zhang Y, Pan Z, Geng S, Jiao X. 2021. WbaP is required for swarm motility and intramacrophage multiplication of *Salmonella* Enteritidis spiC mutant by glucose use ability. *Microbiol Res* 245:126686. <https://doi:10.1016/j.micres.2020.126686>.

Comment 3: With respect to manuscript preparation, the figure axis labels are inconsistent and/or incorrect, and the figure legends require more details.

Response: Figures have been modified, and the details can be seen below(Specific comments).

Comment 4: A control experiment without continuous gentamicin would be useful to rule out the possibility that the compounds screened simply altered macrophage membrane permeability and hence allowed gentamicin entry.

Response: Operation procedure of blank control experiment has been added. "...replaced by 10 µg/mL gentamicin alone or combined with the compound to be screened from the compound library at a concentration of 50 µM, at the same time, no drug treatment was used as blank control.

Specific comments:

Comment 1: "Intracellular bacteria/CFU" in multiple figures when it should be "Intracellular bacteria (CFU)/macrophage" (or better yet, just "CFU/macrophage", with specifics including bacterial strain and macrophage cell line clearly noted in Figure legend);

Response: Figures have been modified, and the details can be seen below;

Comment 2: Confusing, reword: "...antibiotic classes including aminoglycosides penetrate the eukaryote cells, including macrophages (17,18), poorly, the main cell involved as the host for multiplication of the *Salmonella* infections (19,20)."

Response: The sentence has been changed to "antibiotic classes including aminoglycosides poorly penetrate the eukaryote cells, including macrophages (17,18), which is the main cell involved as the host for multiplication of the *Salmonella* infections (19,20).";

Comment 3: "After the first-round screen of 66 compounds..." Insert "(Appendix)". (However, this should probably be "Supplementary Table 1" or the like rather than "Appendix")

Response: "Appendix" has been changed to "Supplementary Table 1" ;

Comment 4: "13 showed the numbers of SE C50041 Δ *spiC* recovered from the cells of less than 10×10^5 CFU, and one of these, E7, was selected in contrast to the drug-free control where 12.5×10^6 CFU were recovered (Fig. 2A)." These numbers do not appear to match those in Fig. 2A. Specifically, the no drug control yielded ca. 125 CFU/5 x 10E5 RAW cells, so to increase this 10E5 times to 12.5×10^6 CFU would require 5×10^{10} RAW, an unlikely high number.

Response: The sentence in the manuscript was modified to "13 compounds including E7 showed the numbers of SE C50041 Δ *spiC* recovered from macrophages with less than 10×10^5 CFU, were selected in contrast to the drug-free control where 12.5×10^6 CFU were recovered (Fig. 2A). Considerable variation in the bacterial counts were observed with some compounds increasing the recovery of viable bacteria."; Fig. 2A(0 to 200) has been recount and redrawn (see Comment 10 for details).

Comment 5: "E7 showed no cytotoxicity at all at the concentration tested of [? insert number] μ M."

Response: The sentence has been added to "E7 showed no cytotoxicity at all at the concentration tested of 25 μ M." ;

Comment 6: "S. Pullorum strain SPS06004 was killed completely by Nifuratel at 100 μ M whereas complete killing was not obtained at this concentration for the S. Typhimurium or S. Dublin strains". Dissapointedly low activity then with all but SE C50041 Δ *spiC*.

Comment 7: "Figure 5. Nifuratel showed potent activity against intramacrophage SE C50041, STm SL1344, SD Sdu189, SP S06004." No, since as noted in text, "potent" activity only obtained versus SE C50041 mutant.

Response: Comment6 and Comment7 were same question about drug effect of Nifuratel. There was "SE C50041", not "SE C50041 Δ *spiC*", which has been modified in line 100. Nifuratel is effective to tested *Salmonella* strains, but the degree is different because of other unknown reasons, which need further study. Nifuratel is effective not only to intramacrophage *Salmonella*, but also to intramacrophage *Shigella* L9 and *Listeria monocytogenes* EGD-e; The legend of Fig.4 and Fig.5 has been changed to "The ability of NIF against intracellular..."

Comment 8: "...mice killed 48 h after oral infection with the SE strain and administered Nifuratel..." Indicate dose.

Response: The sentence has been changed to "The viable numbers of SE C50041 in the liver and spleen of 5 mice killed 48 h after oral infection with 1.0×10^7 CFU, the LD₁₀₀, of SE strain and administered Nifuratel (20 mg/kg) in DMSO 12 h later are shown in Figure 8A and B";

Comment 9: "The P values for each Nifuratel concentration and strain are shown." Not needed here; delete.

Response: The sentence of "The P values for each Nifuratel concentration and strain are shown." has been **deleted**;

Comment 10: Fig 2A y axis label: Add "RAW264.7" after "5 x 10E5" (to reduce length, "Intracellular bacterial load" can be deleted).

Response: The y axis label of Fig. 2A (0 to 200) was the number of colonies diluted to 10⁵, which has been **recount** and **changed** to "Bacterial loads (CFU) in Macrophage". and Fig.2A has been **redawn**(A2、 F5 and F6 were added) ;

Comment 11: Fig. 2B y axis label: Change to "Cell viability and cytotoxicity (%)"

Response: Fig. 2B has been **redrawn** and **divided into two figures** "Cell viability (%)" and "Cytotoxicity (%)"; And the legend has been changed as follows.

Figure 2. Screen of drugs against intracellular *Salmonella*. (A) Loads (CFU) of SE C50041Δ*spiC* in RAW264.7 after treatment with 66 drugs. (B) Viable cell ratio and (C) cytotoxicity after treatment with 28 drugs.

Comment 12: Fig. 2B x axis label: Correct spelling of "Drug" ?

Response: We are sorry for bothering you with this kind of mistakes that we should have avoided. In Fig 2B, "Durg" has been **changed** to "Drug";

Comment 13: Fig. 3A y axis label for graph on left: Why is this axis (10E0 to 10E8) so different than in Fig. 2A (0 to 200)? Use consistent and correct label for this axis.

Response: In the Fig. 3A (left), y axis label has been **changed** to "Bacterial loads

(CFU) in Macrophage". Fig. 2A(0 to 200) has been recount and redrawn (see Comment 10 for details).

Comment 14: Fig. 3A legend: Descriptions needed for graphs in center and on right..

Response: The details of y axis label for Fig.3(middle(LDH), right(MTT)) has been added; and the legend of Fig.3A has been modified as follows.

Figure 3. The ability of NIF against intracellular *Salmonella* Enteritidis without cytotoxicity on the host cells. (A) Loads (CFU) of SE C50041Δ*spiC* in RAW264.7, (B) LDH and (C) MTT result of RAW264.7, (D) RAW264.7 morphology after NIF treatment at different concentrations.

Comment 15: Fig. 4, Fig.5 y axis: Again, why this axis and Fig. 2A axis so different? Also again, correct y axis label and make it consistent with Fig. 2A and 3A.

Response: The y axis label of Fig. 2A (0 to 200) was the number of colonies diluted to 10⁵ which has been recount and redrawn(see Comment 10 for details). The y axis label in Fig. 4 has been changed to "*Salmonella* loads (CFU) in Macrophage", and axis of Fig. 5 has been changed to "*Bacterial* loads (CFU) in Macrophage" ;

Comment 16: Fig. 5 legend: Specify which macrophage used. Insert "*Salmonella* strains" before SE C50041.

Response: The legend of Fig.5 has been changed to "*The ability of NIF against intracellular different Salmonella serotypes.*"; and the "*Salmonella* strains" was inserted before SE C50041 in Fig.4;

Comment 17: "Subsequent experiments showed that Nifuratel was highly active against selected serovars of *Salmonella* enterica." As noted above, "highly active" was only observed for SE C50041Δ*spiC*;

Response: The "SE C50041Δ*spiC*"(line 95,100) in this manuscript should be

modified to "SE C50041". Nifuratel is effective to tested *Salmonella* strains, but the degree is different because of other unknown reasons, which need further study. Nifuratel is effective not only to intramacrophage *Salmonella*, but also to intramacrophage *Shigella* L9 and *Listeria monocytogenes* EGD-e;

Comment 18: "Antibacterial activity, highly likely to be bactericidal..." Speculative since not tested - delete.

Response: The sentence has been deleted;

Comment 19: "It should not be surprising that some new antibiotics and other less conventional chemotherapies are found to be more effective than older, standard antibiotics (27). showed that highly lytic bacteriophages could be more effective than single doses of streptomycin in E. coli-infected mice and equally effective to eight doses." Vague and overly general - delete.

Response: The sentence has been deleted;

Comment 20: "Nifuratel is a nitrofurantoin derivative...It has also been given vaginally." References needed for this entire paragraph.

Response: The reference (Zheng, et al, 2017) has been added;

Reference: 38. Zheng H, Hong H, Zhang L, Cai X, Hu M, Cai Y, Zhou B, Lin J, Zhao C, Hu W. 2017. Nifuratel, a novel STAT3 inhibitor with potent activity against human gastric cancer cells. *Cancer Manag Res* 9:565-572

Comment 21: "Nifuratel is a good potential drug for the first-line treatment of bacterial vaginosis (35,36). It is active in vitro against the pool of bacteria recognized to cause bacterial vaginosis and, at the same time, it does not affect the normal the flora of lactobacilli (35). It is also active against other bacterial species (37,38)." Redundant with text above - delete or reduce considerably.

Response: The sentence has been reduced to "It has also been shown to have low toxicity (34). Nifuratel is a good potential drug for the first-line treatment of bacterial vaginosis (35,36). It is active *in vitro* against the pool of bacteria recognized to cause bacterial vaginosis and, at the same time, it does not affect the normal the flora of lactobacilli (35).";

Comment 22: "Our previous studies have shown that S. Enteritidis (SE) C50041Δ*spiC* had an increased ability to proliferate and persist in cultured macrophages in comparison with the wild-type strain (26). On this basis, we selected this strain for screening in RAW264.7"

Response: The sentence has been modified and moved from Materials and Methods to beginning of Results;

"The screen model for drugs against *Salmonella* in macrophage

In previous study, *S. Enteritidis* (SE) C50041 Δ *spiC* with increased proliferation ability could keep longer-term persistence in cultured macrophages in comparison with the wild-type strain (26), at 20 h post of infection, C50041 Δ *spiC* loads in macrophages by MOI=100 were still enough, but C50041 almost could not recovered, and so on this strain was selected in the study to facilitate the analysis of the drug effect at 20 h poi;"

Comment 23: "...replaced by 10 μ g/mL gentamicin alone or combined with the compound to be screened..." A control experiment without this continuous gentamicin would be useful to rule out the possibility that the compounds screened simply altered macrophage membrane permeability and hence allowed gentamicin entry.

Response: Operation procedure of **blank control** experiment has been **added**. "...replaced by 10 μ g/mL gentamicin alone or combined with the compound to be screened from the compound library at a concentration of 50 μ M, **at the same time, no drug treatment was used as blank control.**";

Comment 24: "...analysis for release of lactate dehydrogenase. Absorbance was measured at 490 nm after mixing 120 μ L cell culture supernatant with 60 μ l LDH assay working solution..." Reference, or reagent source, or more details needed.

Response: The **reference** (Schumann, et al, 2002) has been **added**;

Reference: 46. Schumann G, Bonora R, Ceriotti F, Clerc-Renaud P, Ferrero CA, Férard G, Franck PF, Gella FJ, Hoelzel W, Jørgensen PJ, Kanno T, Kessner A, Klauke R, Kristiansen N, Lessinger JM, Linsinger TP, Misaki H, Panteghini M, Pauwels J, Schimmel HG, Vialle A, Weidemann G, Siekmann L. 2002. IFCC primary reference procedures for the measurement of catalytic activity concentrations of enzymes at 37 degrees C. Part 3. Reference procedure for the measurement of catalytic concentration of lactate dehydrogenase. Clin Chem Lab Med 40(6):643-648

Comment 25: "Twelve hours later Nifuratel (20 mg/kg) in DMSO was also administered by the same route to the mice in group 3" According to Results, this was **24 h** later ("administered Nifuratel **24h** later")

Response: The sentence of "administered Nifuratel 24h later" has been **changed** to "administered Nifuratel **12 h** later";

Responses to Staff :

1. Point-by-point responses to the issues raised by the reviewers has been done in a file named "Response to Reviewers".

Response: Point-by-point responses have been done in a file named "Response to Reviewers".

2. A compare copy of the manuscript (without figures) as a "Marked-Up Manuscript" file has been uploaded.

Response: The "Marked-Up Manuscript (without figures)" file has been done and uploaded.

3. Each figure has been uploaded as a separate file, and any multipanel figures has been assembled into one file.

Response: Each figure has been modified and uploaded as required.

4. Manuscript: A .DOC version of the revised manuscript has been done.

Response: The revised manuscript has been done and in doc format.

5. Figures: Editable, high-resolution, individual figure files are required at revision, TIFF or EPS files are preferred.

Response: The Figures were saved as TIF as required.

Response to Reviewer

Dear Reviewer:

Thank you so much for your careful check. We feel sorry for the inconvenience brought to the reviewer and have revised the manuscript.

1) Please clarify if the supplemental material is intended for publication. Currently your answer to the supplemental material question in eJP is "No," but supplemental material is uploaded and cited in the text.

- If it is for review only, please delete in-text citations or change to "data not shown." Also, change the file type to Miscellaneous.
- If it is intended for online publication, please change your supplemental material answer in eJP to "Yes."

Response: The "Supplementary Table 1" is **not** intended for publication, and has been deleted.

2) Also, your supplemental figure image files are labeled inconsistently. If these are intended to publish, this needs to also be corrected.

Response: The label of figure image files have been **corrected** and **checked** again carefully.

July 26, 2023

Prof. Shizhong Geng
Yangzhou University
Wenhui east road 48#
Yangzhou, Jiangsu 225009
China

Re: Spectrum05147-22R1 (**Nifuratel reduces *Salmonella* survival in macrophages by extracellular and intracellular antibacterial activity**)

Dear Prof. Shizhong Geng:

Your manuscript has been accepted, and I am forwarding it to the ASM Journals Department for publication. You will be notified when your proofs are ready to be viewed.

Sincerely,

Cezar Khursigara
Editor, Microbiology Spectrum
